# Factors Associated with Variability in Pulse Wave Transit Time Using Pulse Oximetry: A Retrospective Study

**DOI:** 10.3390/jcm11143963

**Published:** 2022-07-07

**Authors:** Hilmanda Budiman, Ryo Wakita, Takaya Ito, Shigeru Maeda

**Affiliations:** Department of Dental Anesthesiology and Orofacial Pain Management, Tokyo Medical and Dental University, Yushima, Bunkyo-ku 1-5-45, Tokyo 113-8549, Japan; hilmanda.dds@gmail.com (H.B.); gabacho6333@gmail.com (T.I.); maedas.daop@tmd.ac.jp (S.M.)

**Keywords:** pulse wave transit time, ambulatory monitoring, hemodynamic monitoring, local anesthetics

## Abstract

Pulse wave transit time (PWTT) is the time difference between the occurrence of an R-wave on an electrocardiogram and the detection of pulsatile signals on a pulse oximeter, which reflects changes in blood pressure (BP) corresponding to the vessel wall compliance. However, the factors affecting PWTT variability have not been determined. Thus, we investigated the BP changes associated with variations in PWTT and identified the clinical characteristics associated with these variations. Data related to 605 cases of dental procedures performed under intravenous conscious sedation from April 2020 to November 2021 were collected, and 485 cases were enrolled. Heart rate, systolic blood pressure before and after local anesthesia (LA) administration, and crest and trough PWTT waves during LA administration were recorded. Thereafter, PWTT variability was calculated; cases were divided into two groups: large PWTT variability (LPV, *n* = 357) and small PWTT variability (SPV, *n* = 128). The index of large PWTT variability could not detect changes in BP. Logistic regression analysis revealed that factors, such as LA use, age, hypertension, and dental treatment phobia were associated with PWTT variability. The use of epinephrine more than 36.25 µg in each LA resulted in PWTT variability of more than 15 ms.

## 1. Introduction

Scheduled intermittent measurement of non-invasive blood pressure (NIBP) is important for detecting hemodynamic changes throughout dental procedures performed under intravenous conscious sedation (IVCS) [1]. Nevertheless, NIBP measurements recorded at specific intervals may not be sufficient to detect short-term changes in blood pressure (BP), which are associated with rapid changes in the vascular properties. Particularly in dental procedures, the use of local anesthetic agents containing epinephrine for infiltration anesthesia in addition to various stimuli involving autonomic responses, such as invasive treatments associated with pain and stress, may induce large BP fluctuations within a short period of time.

Epinephrine is commonly contained in local anesthetic agent for routine dental procedures and a potent vasoconstrictor that stimulates α-adrenergic receptors, resulting in immediate BP variations after local anesthesia (LA) administration. This may not be detected by intermittent NIBP measurement and failure to detect BP fluctuations may increase the risk on patients with underlying cardiovascular disease [2,3]. Anesthesiologists can resolve this issue through manual assessments, which may however be subjective. Frequent measurements may also cause discomfort, trauma, and nerve injury to the patient [4].

Various approaches have been investigated as alternatives for continuous BP monitoring [5]. Pulse wave transit time (PWTT) is one of these, and vital monitoring devices that make use of this parameter are commercially available. It calculates the time between the rise of the photoplethysmography (PPG) waveform and the R-wave of the electrocardiogram by using the changes in finger blood volume determined by the PPG [6]. PWTT is affected by changes in vascular volume, sympathetic nerve activity, and vascular elasticity [3,7]. The pulse wave propagates through the arterial vessel toward the peripheral measurement site, where it appears as a time delay (milliseconds). The length of the PWTT is directly proportional to the BP [4]. When blood volume is high or vessels are constricted, pulse waves travel faster because blood flow reaches the peripheral site with high speed. Variations in PWTT reflect vascular tone increases and subsequent stiffening of the vessel walls [8]. Large variations in PWTT can occur when the vessel wall has a normal ability to constrict under relaxed conditions, especially when it has high elasticity. In contrast, small variations in PWTT might indicate that the vessel wall compliance is limited. This condition can arise from factors such as arteriosclerosis [9]. As a result, the decrease in PWTT is reflected as a steep drop in the graph, showing its changes over time, and the BP increases [10,11]. In this graph, changes in PWTT are indicated by the amplitude width and height. If the vessel wall contracts rapidly, the PWTT drops abruptly; if it contracts severely, the PWTT amplitude increases.

The clinical measurement of PWTT began in recent years. One bedside monitor (Life Scope BSM 3562, Nihon Kohden, Tokyo, Japan) has two settings that automatically trigger NIBP measurement using PWTT; one is when PWTT varies by more than 15 ms, which is assumed to be a change in systolic blood pressure (SBP) of >20 mmHg. The second was when the estimated SBP calculated from this PWTT change exceeded the SBP alarm setpoint for more than 8 s. NIBP measurements are triggered when both conditions are satisfied [12].

Previous studies on the direct relationship between PWTT and BP are often inadequate because of differences in the clinical characteristics of the subjects and the small sample sizes [6,10]. Meanwhile, an investigation of PWTT and BP in newborns showed the ability of PWTT to continuously detect BP changes [11]. To our knowledge, no study has examined the association between changes in BP and variations in PWTT during dental procedures. Therefore, the aim of this study was to investigate whether BP changes can be detected early from the variability in PWTT and the clinical characteristics related to these variations during dental treatment.

## 2. Materials and Methods

### 2.1. Participants and Data Sources

We conducted a retrospective study on the changes in PWTT following infiltration anesthesia in adult patients with American Society of Anesthesiologists (ASA) physical status score I–III (I, normal healthy patient; II, indicating patient with mild systemic disease; and III, indicating patient with severe systemic disease with no constant threat to life), who received dental treatment under IVCS between April 2020 and November 2021 in our dental anesthesiology clinic. The ethics committee of our university (No. D2021-017) approved the study procedures for sample collection and analysis. In accordance with the Ethical Guidelines for Medical and Health Research Involving Human Subjects, we substituted the consent form with disclosure of the study details on the hospital website and posting. Patients’ background data were acquired from medical records, vital signs including PWTT were collected from anesthesia charts, and bedside monitors at the Dental Anesthesiology Clinic of the TMDU Hospital, respectively. The exclusion patient’s criteria were as follows: less than 20 years old, history of frequent arrhythmia or pacemaker use, use of LA without epinephrine, and no LA use.

After standard monitoring was performed, including an electrocardiogram, measurement of NIBP every 5 min, and a pulse oximeter on the index finger of the hand, a peripheral venous catheter was placed in the patient’s right arm. In our institution, midazolam 0.03–0.04 mg/kg was administered as a single dose at the usual induction, followed by continuous propofol at 1–4 mg/kg/h, with a target sedation level of 4–5 on the Ramsay sedation scale. Patients received 100% O_2_ inhalation through a nasal catheter at a flow rate of 3 mL/min during the treatment. After confirming stable vital signs and achieving the target sedation level, LA was administered. A 2% lidocaine solution with epinephrine concentration of 1:80,000 (1.8 mL/cartridge) was used as the local anesthetic, and the amount used was decided by the surgeon. Heart rate (HR), SBP, and PWTT were recorded on a bedside monitor from the time of NIBP measurement immediately before LA administration until the next scheduled NIBP measurement, 5 min after completely administering the first dose of LA. The BSM PC Viewer, version 0.04 (Nihon Kohden, Tokyo, Japan) was used to record and extract HR, SBP, and PWTT.

### 2.2. Clinical Characteristics

We gathered the following patient data from the available medical records and anesthesia charts: sex; age; weight; height; body mass index (BMI) at the time of treatment; obesity classified as BMI ≥ 25 [13]; ASA physical status score; dose of epinephrine (µg) in LA solution that was used during the recorded period; preexisting patient history of systemic diseases or mental disorders; patients with history of smoking cigarettes at least 1 d in the past month during medical examination were classified as smokers [14]; and history of alcohol intake in medical records before the treatment classified as alcohol consumption. We recorded the entire history, rather than just a representative one, because patients could have had multiple medical conditions.

### 2.3. Variables from Recorded Segments

We established the baseline heart rate (HR_BL_) and baseline systolic blood pressure (SBP_BL_) at the time of NIBP measurement immediately before LA administration. Maximum heart rate (HR_LA_) and maximum systolic blood pressure (SBP_LA_) were recorded at the time of NIBP measurement more than 5 min after finishing LA administration. ΔHR and ΔSBP were obtained from the differences between HR_BL_ and HR_LA_, and SBP_BL_ and SBP_LA_, respectively (Figure 1). PWTT variability (ΔPWTT) was calculated as the difference between crest PWTT wave (PWTT_MAX_) and trough PWTT wave (PWTT_MIN_) during the LA administration. The patients were then divided into two groups: a large variation group with ΔPWTT > 15 (LPV) and a small variation group with ΔPWTT < 15 (SPV).

### 2.4. Statistical Analysis

The normality of data distribution on clinical characteristics and variables was examined using the Shapiro–Wilk and Kolmogorov–Smirnov tests. Normally distributed variables were expressed as the mean ± standard deviation. Non-normally distributed data were presented as medians with interquartile range, minimum, and maximum. The distribution of clinical characteristics and variables from recorded segments was compared between the LPV and SPV groups using *t*-tests or Mann–Whitney U tests for continuous variables and chi-square tests or Fisher’s exact tests for categorical data, as appropriate. To evaluate the primary outcome, i.e., the relationship between LPV and ΔSBP, the Mann–Whitney U test was used. Logistic regression was used to determine the associations between clinical characteristics and the LPV group. All statistical analyses were performed using SPSS Statistics for Windows, version 27.0 (IBM Corp., Armonk, NY, USA). Statistical significance was set at *p* < 0.05.

## 3. Results

### 3.1. Primary Outcome and Summary of Inclusions

Six-hundred-and-five patients who received dental treatment under IVCS were enrolled. Seventy-one patients were excluded based on the exclusion criteria. A total of 534 anesthesia records were reviewed, and in 49 cases, either the PWTT was interrupted or some data were not recorded. Therefore, 485 cases were included in the statistical analyses. Figure 2 presents a flow chart of the case selection process. The included cases were deemed as fit for the analyses, with 357 cases (73.6%) grouped into the LPV group and 128 cases (26.4%) into the SPV group. The average age was 49 ± 17 years old with a sex distribution of 61.2% female and 38.8% male with regard to the total population in this study. There were significant differences in sex, ASA scoring, age, weight, height, BMI, LA, and PWTT_MAX_ between the two groups. In the SPV group, the mean age of the patients was higher. The use of epinephrine in local anesthetic agent was higher in the LPV group. (Table 1).

The median ΔPWTT was distinctively higher in the LPV group, and there was no significant difference between the changes in SBP and PWTT variability over 15 ms, which was the primary outcome of this study. ΔHR was also not significantly different from PWTT variation. (Table 2).

Regarding variables from the medical histories, there were significant differences in hypertension, cardiac disease, and hepatitis c between the two groups. Hypertension and obesity are more prevalent in the LPV group (Table 3).

### 3.2. Results from Logistic Regression Analysis

Independent risk factors for LPV were age (odds ratio [OR] = 0.974, 95% confidence interval [CI] = 0.96–0.99), LA cartridge (OR = 2.417, 95% CI = 1.7–3.2), hypertension (OR = 1.896, 95% CI = 1.07–3.5), and dental treatment phobia (OR = 1.74, 95% CI = 1.07–3.03) (Table 4). All clinical characteristics were included as variables in logistic regression analysis using the backward stepwise likelihood method. The probability of exhibiting LPV increased 2.4-fold for every additional LA cartridge used. Patients with hypertension and dental treatment phobia have a likelihood of more than 1.5 times to establish LPV. However, increasing age was found to be less likely to experience LPV.

#### Supplementary Analysis

Based on the result that PWTT varies significantly in response to increased LA use, which means increased epinephrine use, we decided to perform an additional analysis of the cut-off values of epinephrine dose at which the PWTT changed by >15 ms, using the receiver operating characteristic curve analysis and Youden index. The results showed that at a dose of 36.25 μg of epinephrine, the PWTT began to fluctuate for >15 ms (Youden index: 1.281, sensitivity: 69.5%) (Figure 3).

As a post-hoc analysis, the relationship between ΔPWTT and ΔSBP was determined using Spearman’s correlation. This analysis showed a statistically significant positive correlation between PWTT variability and SBP change (r_s_ = 0.196, *p* < 0.001) (Appendix A).

Furthermore, we used a multiple regression model to confirm the association of ΔPWTT with clinical characteristics, using the same data to confirm the factors influencing ΔPWTT obtained from the logistic regression analysis. Age, LA cartridge, respiratory disease, and phobia were statistically significant factors for PWTT variability (Appendix A).

## 4. Discussion

The main outcome of our study was the early detection of changes in BP because of large changes in PWTT. The vital monitor used in this study triggers NIBP measurement in situations when a variation in PWTT of >15 ms is detected and the estimated SBP (a PWTT variation of >15 ms is defined as a 20 mmHg increase in SBP) remains above the alarm limit for more than 8 s [12]. Although we used only this PWTT variation (PWTT variability > 15 ms) as our threshold, it was not sufficient to detect BP changes in this study. A previous study found that PWTT was related to stroke volume (SV) but a change in BP was not related to SV [15]. Therefore, a large PWTT variability that occurs under stimulation does not always reflect BP changes. Another possibility is that because NIBP measurement takes at least 10 s, PWTT changes > 15 ms may reflect NIBP changes only if the change lasts for a certain period of time. However, the association between PWTT and BP changes exists because we found significant positive correlation between PWTT variability and SBP changes from our supplementary analysis (data was not presented, available as Appendix A). Even the slightest PWTT change might reflect changes in SBP; therefore, smaller PWTT thresholds can be considered for detecting BP fluctuations. However, a small change might be an excessive trigger, and it is necessary to reconsider whether these BP fluctuations are clinically meaningful.

We conducted a logistic regression analysis to identify factors related to LPV during dental procedures performed under IVCS and found that increased LA use, younger age, hypertension, and phobia were independent clinical characteristics. As a validation, multiple regression analysis was performed, and the results were similar to those of logistic regression analysis. These results raised the certainty that ΔPWTT increased with the amount of LA that was administered and presence of dental treatment phobia and decreased with factors such as advanced age, which means that these factors were related to the vessel wall compliance. PWTT reflects vessel wall compliance and is influenced by vascular resistance but not cardiac output [16]. Briefly, epinephrine in local anesthetics changes the somatic vascular resistance via α-action, resulting in increased vascular resistance and shortened PWTT. Moreover, epinephrine increases the HR and SV of the heart, leading to an increase in the cardiac output because of the action on β-adrenergic receptors. However, β-receptor stimulation with low doses of epinephrine dilate arterial vessels in the visceral and skeletal muscles, resulting in lower vascular resistance [17]. This suggests that the increase in BP because of α-action can be attenuated by the β-action. Therefore, the administration of local anesthetics may cause changes in PWTT but may not reflect changes in BP. This may be one of the reasons why we did not find a statistically significant difference between the LPV and ΔSBP in our study.

Vessel wall contraction affects systemic vascular resistance, which is one of the factors that defines BP [18]. However, variations in PWTT associated with local anesthetic injections may reflect not only peripheral vasoconstriction but also SV, i.e., the entire hemodynamic change. Therefore, an additional analysis was performed to determine the cut-off dose of epinephrine, which revealed that a dose of 36.25 µg of epinephrine was the cut-off value. One milliliter of 2% lidocaine with 1:80,000 epinephrine solution contains 12.5 µg of epinephrine, which is approximately equivalent to three 1.8 mL LA cartridges at this cut-off value. Previous studies have indicated that 18–36 μg of epinephrine may have little clinical change in most patients, including those with hypertension or other cardiac diseases, but ≥36 μg epinephrine may have some effect on the circulating system by significantly increasing SV and cardiac output [19,20,21]. An increase in SV also increases pulse wave propagation; therefore, a large PWTT variability is expected to occur [15]. LPV may not reflect BP changes but may reflect hemodynamic changes that are not reflected in BP.

The incidence of LPV increased in hypertension. A previous study reported that pulse pressure and PWTT are associated with vasoconstriction in adults [22]. Aortic compliance in hypertensive patients decreases with age, resulting in an increase in pulse pressure and SBP, whereas peripheral vessel wall compliance remains relatively unchanged [23]. A higher pulse pressure may lead to a more pronounced anterior pulse pressure wave toward peripheral sites, which is larger in amplitude and steeper in pulse wave elevation, indicating a steeper increase in PWTT [24]. Moreover, under stimuli such as epinephrine, hypertension does not alter peripheral compliance, but decreases the aortic wall compliance. Therefore, hypertension may show large PWTT variability.

Dental treatment phobia in this study was defined as patients who feel fear or anxiety of undergoing dental treatment, which patients usually do not feel. A psychological experiment reported a strong correlation between PWTT and stress [25]. Patients with dental phobia may exhibit increased respiratory rate, HR, vasoconstriction, and BP [26]. These reactions are caused by the activation of the sympathetic nervous system and hormone release, such as epinephrine, which increases PWTT variability. Endogenous release of catecholamines, specifically epinephrine, increases 20–40 times under stress, including anxiety, compared with normal conditions [27]. Circulating epinephrine has been implicated as a contributor to embedding non-conscious emotional memories of fearful or threatening events in the amygdala [28]. When emotional memory is embedded too strongly in the amygdala, it can produce a heightened fear response to external events that is out of proportion to the actual nature of the problems, such as patients with a history of very painful dental treatment developing dental phobia [29]. Since emotional memories stored in the amygdala cannot be consciously controlled, it can be difficult to eradicate or regulate events perceived as stressors. This memory triggers a chronic fear response and places the brain, body, and mind in a constant state of alertness, resulting in a greater release of epinephrine [30]. Endogenous epinephrine in patients with dental phobia may remain in the body after receiving IVCS, and subsequent administration of LA immediately activates β1 receptors, thereby increasing the intensity of vascular wall contraction [31]. Therefore, large fluctuations in PWTT are likely to occur in patients with dental phobia.

The incidence of LPV is lower in the older population. Repetitive contraction-relaxation with aging is thought to change vessel wall properties, reducing the elastic fibers of the vessel wall [32]. A decrease in vascular wall compliance should result in large PWTT fluctuations during vasoconstrictor stimulation; however, the reason for the opposite result in this study is unclear.

In this study, we found that the threshold of PWTT variations of >15 ms did not reflect changes in NIBP. However, on the basis of the association of ΔPWTT with ΔSBP and the epinephrine cut-off values, it is possible that PWTT variability can detect early hemodynamic changes after the administration of epinephrine-containing LA. Since PWTT is substantially influenced by vessel wall compliance and peripheral arterial resistance, PWTT may reflect fluctuations in not only the blood pressure but also in the circulation following administration of LA. Further, PWTT fluctuations can be observed noninvasively using conventional vital sign measurements. PWTT measurement remains a potential simple and affordable alternative to systemic vascular resistance measurements in patients with systemic diseases who receive dental treatment [33].

This study has certain limitations. The main limitation of this study was its retrospective nature, which makes it susceptible to selection bias. Although we attempted to control for confounding factors by collecting data in a comprehensive manner, there may still be unknown biases, such as differences in surgical management, consciousness level, or sedative agents used. Second, there is a disparity in the interval between LA administration and BP measurement in our study. In addition, we did not classify the arterial stiffness or identify related factors that may have significantly affected the PWTT. Future prospective studies must consider these factors and the measurement of parameters, including PWTT and vital signs, at the same time points.

## Figures and Tables

**Figure 1 jcm-11-03963-f001:**
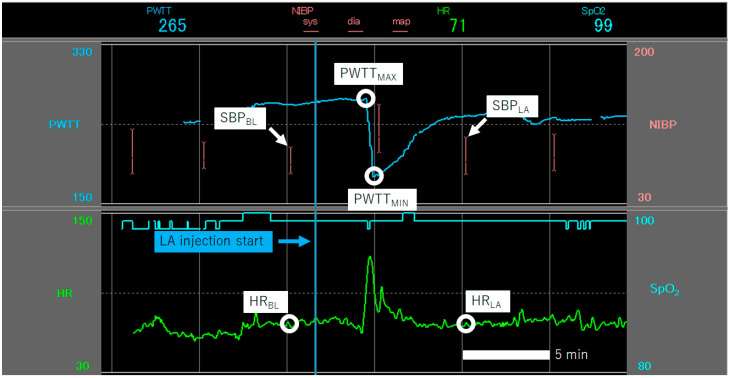
Baseline HR (HR_BL_) and baseline SBP (SBP_BL_) immediately before local anesthesia; heart rate after local anesthesia (HR_LA_); and systolic blood pressure (SBP_LA_) were also recorded at 5 min after local anesthesia. Value for ΔPWTT. Crest PWTT wave (PWTT_MAX_) value; trough PWTT wave (PWTT_MIN_) value. Unit PWTT in milliseconds (msec), SBP in mmHg, HR in beats per minute (bpm). Abbreviations: HR = heart rate; SBP = systolic blood pressure; PWTT = pulse wave transit time.

**Figure 2 jcm-11-03963-f002:**
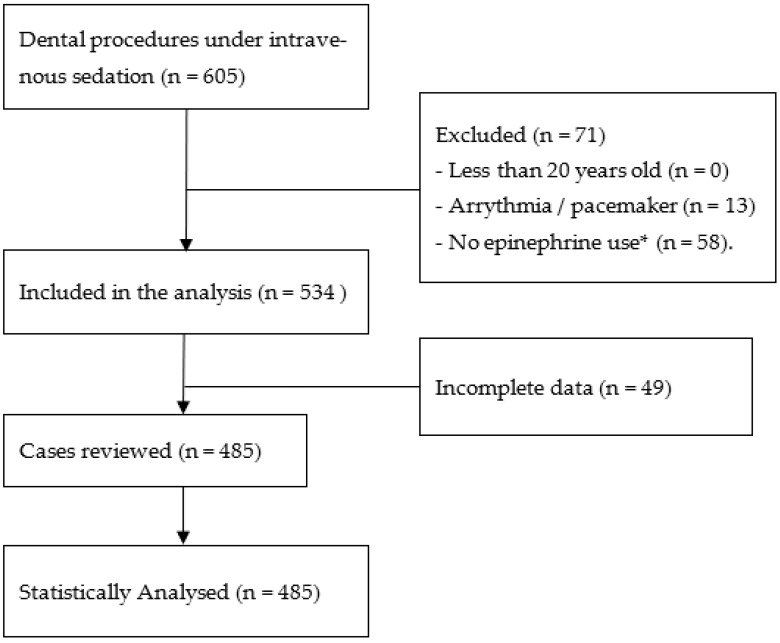
Flow chart of the case selection criteria. * Cases did not receive infiltration anesthesia and infiltration anesthesia without epinephrine.

**Figure 3 jcm-11-03963-f003:**
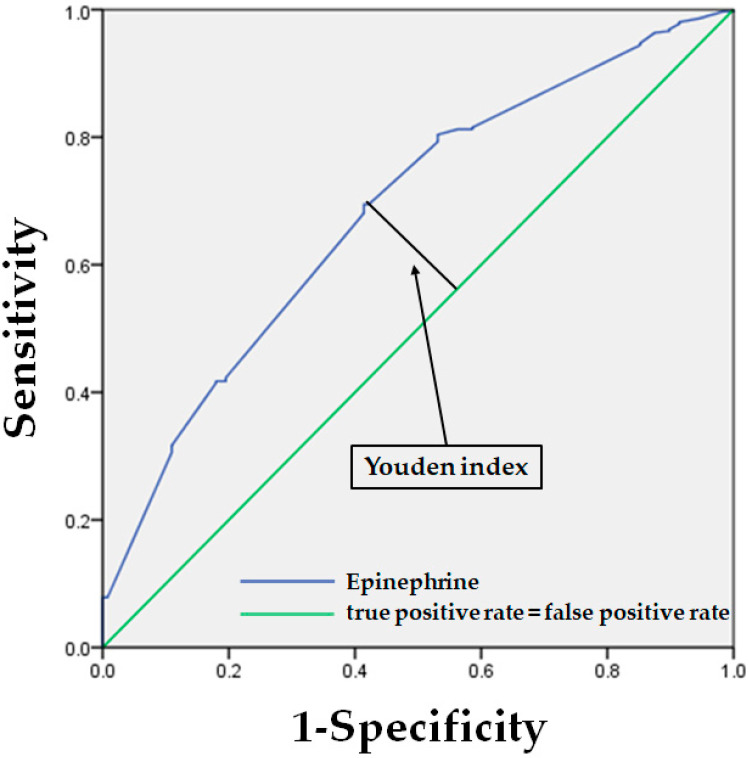
Receiver operating characteristic curves representing epinephrine cut-off value with Youden index.

**Table 1 jcm-11-03963-t001:** Descriptive and univariate analysis of variables from recorded segments and clinical characteristics from overall reviewed cases.

Variables	Descriptive Analysis	Univariate Analysis		
	Overall Cases (*n* = 485)	LPV (*n* = 357)	SPV (*n* = 128)	*p* Value
Age (year)	49 (17)	48.1 (17)	52.3 (17)	0.018 *
Gender				<0.003 *
Male	188 (38.8%)	122 (64.9%)	66 (35.1%)	
Female	297 (61.2%)	235 (79.1%)	62 (20.9%)	
ASA				0.008 *
I	225 (46.4%)	180 (37.1%)	45 (9.3%)	
II	256 (52.8%)	175 (36.1%)	81 (16.7%)	
III	4 (0.8%)	2 (0.4%)	2 (0.4%)	
Weight (kg)	60.5 (13)	58.5 (13)	62.5 (13)	0.004 *
Height (m)	1.6 (0.1)	1.62 (0.1)	1.64 (0.1)	0.034 *
Body mass index (kg/m^2^)	22.5 (4)	22 (4)	23 (4)	0.019 *
Midazolam (mg)	2 (0–5.9)	2 (0–5.9)	2 (0–5)	0.973
Epinephrine use in LA (µg)	43 (20)	49.9 (23)	35.5 (17)	<0.001 *
PWTT_MAX_ (ms)	227 (168–301)	228 (168–301)	220 (179–269)	0.048 *
PWTT_MIN_ (ms)	202 (146–270)	198 (146–270)	212 (168–265)	0.551
SBP_BL_ (mmHg)	115 (85–161)	115 (85–160)	117 (86–182)	0.197
SBP_LA_ (mmHg)	116 (79–182)	117 (86–182)	116 (79–173)	0.536
HR_BL_ (bpm)	72 (42–122)	72 (42–122)	72 (48–102)	0.896
HR_LA_ (bpm)	78 (45–128)	78 (45–128)	77 (45–112)	0.766

* Indicates statistically significant variables *p* < 0.05. Data shown as mean ± standard deviation, counts (percent), or median (interquartile range; minimum and maximum). Abbreviations: ASA = American Society of Anesthesiologists; HR_BL_ = baseline heart rate; HR_LA_ = post local anesthesia heart rate; LA = local anesthesia; LPV = large pulse wave transit time variability group; PWTT = pulse wave transit time; PWTT_MAX_ = crest pulse wave transit time; PWTT_MIN_ = trough pulse wave transit time; SBP_BL_ = baseline systolic blood pressure; SBP_LA_ = post local anesthesia systolic blood pressure; SPV = small pulse wave transit time variability group.

**Table 2 jcm-11-03963-t002:** Mann–Whitney test of LPV toward ΔSBP and ΔHR.

Variables	LPV	SPV	*p* Value
	(*n* = 357)	(*n* = 128)	
ΔPWTT (ms)	31 ((−15)–(−88))	10 ((−14)–6)	<0.001 *
ΔSBP (mmHg)	0 ((−39)–63)	−1 ((−43)–32)	0.072
ΔHR (bpm)	5 ((−36)–180)	3 ((−14)–179)	0.626

* Indicates statistically significant variables *p* < 0.05. Data shown as median (interquartile range; minimum and maximum). Abbreviations: HR = heart rate; LPV = large pulse wave transit time variability group; PWTT = pulse wave transit time; SBP = systolic blood pressure; SPV = small pulse wave transit time variability group.

**Table 3 jcm-11-03963-t003:** Descriptive and univariate analysis of medical history variables from overall reviewed cases.

Variables ^1^	Descriptive Analysis	Univariate Analysis		
	Overall Cases (*n* = 485)	LPV (*n* = 357)	SPV (*n* = 128)	*p* Value
Hypertension	128 (26.4%)	77 (60.2%)	51 (39.8%)	<0.001 *
Cardiac disease	21 (4.3%)	11 (52.4%)	10 (47.6%)	0.024 *
Hepatitis C	2 (0.4%)	0 (0%)	2 (1.6%)	0.018 *
Cerebral hemorrhage	3 (0.6%)	3 (100%)	0 (0%)	0.298
Cerebral infarction	3 (0.6%)	2 (66.7%)	1 (33.3%)	0.784
Epilepsy	10 (2.1%)	7 (70%)	3 (30%)	0.794
Depression	15 (3.1%)	11 (73.3%)	4 (26.7%)	0.98
Other mental illness	44 (9.1%)	32 (72.7%)	12 (27.3%)	0.69
Asthma	47 (9.7%)	36 (76.6%)	11 (23.4%)	0.619
Emphysema	2 (0.4%)	1 (50%)	1 (50%)	0.448
Other respiratory disease	18 (3.7%)	14 (77.8%)	4 (22.2%)	0.683
Liver/biliary tract disease	1 (0.2%)	1 (100%)	0 (0%)	0.549
Kidney/urinary disease	10 (2.1%)	7 (70%)	3 (30%)	0.797
Obesity ^2^	62 (12.7%)	39 (62.9%)	23 (37.1%)	0.041 *
Diabetes	21 (4.3%)	12 (57.2%)	9 (42.8%)	0.08
Other metabolic disease	10 (2.1%)	9 (90%)	1 (10%)	0.235
Gynecological disease	7 (1.4%)	6 (85.7%)	1 (14.3%)	0.464
Dental treatment phobia	214 (44.1%)	156 (72.9%)	58 (27.1%)	0.711
Glaucoma	20 (4.1%)	13 (65%)	7 (35%)	0.372
Hyperactive pharyngeal reflex	46 (9.5%)	32 (69.6%)	14 (30.4%)	0.513
Hyperlipidemia	45 (9.3%)	30 (66.7%)	15 (33.3%)	0.267
Autoimmune disease	13 (2.7%)	12 (92.3%)	1 (7.7%)	0.121
Vasovagal syncope	11 (2.3%)	7 (63.6%)	4 (36.4%)	0.448
Alcohol consumption	141 (29.1%)	103 (73%)	38 (27%)	0.833
Smokers	42 (8.6%)	33 (78.6%)	9 (21.4%)	0.458

* Indicates statistically significant variables *p* < 0.05. Data shown as counts (percent). ^1^ Medical history are not mutually exclusive because patients can have more than one criterion. ^2^ body mass index over than 25. Abbreviations: LPV = large pulse wave transit time variability group; SPV = small pulse wave transit time variability group.

**Table 4 jcm-11-03963-t004:** Logistic regression analysis on clinical characteristic toward LPV.

Characteristics	*B* (sd)	Odds Ratio (95% CI)	*p* Value
Age (years)	−0.025 (0.009)	0.974 (0.96–0.99)	0.006 *
Local anesthesia cartridge (1.8 mL)	0.868 (0.158)	2.417 (1.7–3.2)	<0.001 *
Hypertension (Yes)	0.663 (0.303)	1.896 (1.07–3.5)	0.028 *
Dental treatment phobia (Yes)	0.591 (0.264)	1.74 (1.07–3.03)	0.025 *
Constant	−0.614 (1.221)	0.541 (-)	0.615

* Indicates statistically significant variables *p* < 0.05. Abbreviations: HR = heart rate; LPV = large pulse wave transit time variability group.

## Data Availability

The data presented in this study are available on request from the corresponding author.

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
