# Peer review of "Factors Associated with Variability in Pulse Wave Transit Time Using Pulse Oximetry: A Retrospective Study"

_jcm, 2022, doi:10.3390/jcm11143963_

Round 1

Reviewer 1 Report

Dear authors, 

you present a nice physiological clinical study on dental patients in light sedation. Your methods are sufficiently robust. Results are clearly presented and discussed. They reflect the pathophysiology of autonomic nervous system instability in stress situations as in your dental patients with anxiety showing higher variability. 

I have no major comments, only minor ones:

1) State more clearly in the methods that the study was retrospective, that it was approved by a local ethical committee, and that patient informed consent was not required. 

2) It is confusing to point ASA V in the methods section. It leads the reader to feel that you included even ASA V patients that are generally not suitable for office-based procedures/outpatient clinics. 

3) You present results of patent enrollment in the methods section, this should be in the results section. 

4) In the citations you should indicate first 6 authors and et al.

Author Response

Dear Honorable Reviewer,

We are very grateful for your valuable comments and suggestions on our manuscript.

We have made adjustment according to your input.
Please see the attachment.

Best regards.

Reviewer 2 Report

Dear Authors 

The study is very interesting 

Title:

Accurate 

Abstract:

Concise  and appropriate 

Rephrase the below statement:

Additionally, epinephrine dose of >36.25 µg in 20 each LA injection caused PWTT variability of >15 ms.

Keywords: Use MeSH terms 

Introduction:

The second paragraph should be consiced 

The purpose is not very clear 

The literature  should draw the need of the study.

Line 72 add references for Previous studies []

Methods:

The inclusion exclusion criteria is not clear 

what catogery of ASA subjects involved in the study.

What is reference for BMI it varies with population and age 

Results well explained 

Figure 3 X axis and y axis ??

Discussion:

Well written can be improved by discussing the limitations and strengths in eloborative manner.

The clinical relvenc e of the study also  add value to the manuscript.

 Conclusion:

based on objective 

Reference: 

appropriate list 

Author Response

(The authors gave the same response as above.)

Round 2

Reviewer 2 Report

Authors responded to all the queries 

i feel now the paper is in better shape.